# Inflammation and the Association of Vitamin D and Depressive Symptomatology

**DOI:** 10.3390/nu13061972

**Published:** 2021-06-08

**Authors:** Ezgi Dogan-Sander, Roland Mergl, Anja Willenberg, Ronny Baber, Kerstin Wirkner, Steffi G. Riedel-Heller, Susanne Röhr, Frank M. Schmidt, Georg Schomerus, Christian Sander

**Affiliations:** 1Department of Psychiatry and Psychotherapy, University of Leipzig Medical Center, 04103 Leipzig, Germany; F.Schmidt@medizin.uni-leipzig.de (F.M.S.); Georg.Schomerus@medizin.uni-leipzig.de (G.S.); Christian.Sander@medizin.uni-leipzig.de (C.S.); 2Institute of Psychology, Universität der Bundeswehr München, 85577 Neubiberg, Germany; roland.mergl@unibw.de; 3Institute of Laboratory Medicine, Clinical Chemistry and Molecular Diagnostics, University of Leipzig Medical Center, 04103 Leipzig, Germany; Anja.Willenberg@medizin.uni-leipzig.de (A.W.); Ronny.Baber@medizin.uni-leipzig.de (R.B.); 4LIFE—Leipzig Research Center for Civilization Diseases, University of Leipzig, 04103 Leipzig, Germany; kwirkner@life.uni-leipzig.de; 5Institute for Medical Informatics, Statistics and Epidemiology (IMISE), University of Leipzig, 04107 Leipzig, Germany; 6Institute of Social Medicine, Occupational Health and Public Health, University of Leipzig, 04103 Leipzig, Germany; Steffi.Riedel-Heller@medizin.uni-leipzig.de (S.G.R.-H.); Susanne.Roehr@medizin.uni-leipzig.de (S.R.); 7Global Brain Health Institute (GBHI), Trinity College Dublin, D02 PN40 Dublin, Ireland

**Keywords:** depression, vitamin D, inflammation, mediation, moderation, LIFE-Adult-Study

## Abstract

Depression and vitamin D deficiency are major public health problems. The existing literature indicates the complex relationship between depression and vitamin D. The purpose of this study was to examine whether this relationship is moderated or mediated by inflammation. A community sample (*n* = 7162) from the LIFE-Adult-Study was investigated, for whom depressive symptoms were assessed via the German version of CES-D scale and serum 25-hydroxyvitamin D (25(OH)D) levels and inflammatory markers (IL-6 and CRP levels, WBC count) were quantified. Mediation analyses were performed using Hayes’ PROCESS macro and regression analyses were conducted to test moderation effects. There was a significant negative correlation between CES-D and 25(OH)D, and positive associations between inflammatory markers and CES-D scores. Only WBC partially mediated the association between 25(OH)D levels and depressive symptoms both in a simple mediation model (ab: −0.0042) and a model including covariates (ab: −0.0011). None of the inflammatory markers showed a moderation effect on the association between 25(OH)D levels and depressive symptoms. This present work highlighted the complex relationship between vitamin D, depressive symptoms and inflammation. Future studies are needed to examine the effect of vitamin D supplementation on inflammation and depressive symptomatology for causality assessment.

## 1. Introduction

Depression is highly prevalent affecting approximately 350 million people worldwide [1]. Depressive disorders are also considered as a leading cause of disability [2] and associated with increased mortality and health care costs [3]. Although the etiology and underlying pathophysiology have not been fully clarified, several mechanisms have been discussed based on the new research developments. 

In recent years, a growing body of evidence has indicated the possible association between vitamin D and depression. Like depression, vitamin D deficiency is also one of the major public health problems. A recent study investigating vitamin D deficiency in Europe [4] showed that 13% of participants were suffering from vitamin D deficiency (defined as 25-Hydroxyvitamin D (25(OH)D) levels below 30 nmol/L). Although many meta-analyses and systematic reviews indicate that depression may be related to low serum vitamin D concentrations [5,6,7], there are also inconsistent results [8,9]. Still, the majority of the current research literature suggest a role of vitamin D in the pathophysiology of depression. Several potential mechanisms have been proposed: first, the presence and distribution of vitamin D receptors and the enzyme 1 α-hydroxylase (catalyzes conversion of 25(OH)D to 1,25-dihydroxycholecalciferol (1,25(OH)_2_D)) within brain regions such as hypothalamus, prefrontal cortex, substantia nigra, amygdala and thalamus [10] support the hypothesis that vitamin D could affect the brain. Furthermore, the presence of vitamin D receptor in several neurons and microglial cells indicates the role of vitamin D in immune responses in central nervous system. Second, there is evidence on the neuroprotective and neurotrophic effect of vitamin D on brain function [11,12], whereas an imbalance between neurodegenerative and neuroprotective markers had been suggested to play a critical role in the pathophysiology of depression [13]. Third, it has been shown that 1,25(OH)_2_D—the active form of vitamin D in the human body—can transcriptionally activate tryptophan hydroxylase 2 and thereby increase the serotonin synthesis [14], which was found to be altered in depression. Fourth, studies are indicating the modulatory role of vitamin D in adaptive and innate immune systems via multiple mechanisms such as regulating cytokine secretion as well as cell signaling pathways [15,16]. For instance, decreased levels of inflammatory markers were observed after vitamin D treatment in several patient groups [17,18].

In recent years, a large body of evidence attests to a role of the immune system in the pathogenesis of depressive disorders: previous studies have indicated an association between depressive symptoms and low-grade inflammatory response [19,20,21]. Furthermore, a decrease in pro-inflammatory markers was observed in several studies after treatment with anti-depressants (e.g., [21,22]). Moreover, it has been reported that inflammation can be a risk factor for the development of depression [23,24].

There are few studies examining the complex relationship of vitamin D, depression and inflammation [25,26,27,28]. Accortt et al. reported that in African American women (*n* = 91) the association between postpartum depressive symptoms and prenatal 25(OH)D levels were significantly moderated by interleukin 6 (IL-6) and the ratio of IL-6/IL-10 [26]. Still, individual inflammatory markers were not associated with Edinburgh Postnatal Depression Scale scores and prenatal 25(OH)D levels were only marginally associated with postnatal depressive symptoms [26]. Moreover, higher levels of postpartum depressive symptoms were related with lower prenatal serum 25(OH)D levels in women with increased levels of IL-6 and greater IL-6/IL-10 ratios. A test of moderation by C-reactive protein (CRP) was not significant. In another study among Korean employees (*n* = 52228 with almost 86% of participants men) [27], a significant negative association between vitamin D status and depressive symptoms was found, but the CRP levels were not associated with vitamin D levels and a positively significant association with depressive symptoms was lost after excluding datasets with abnormal CRP levels (>10 mg/L). Grudet et al. reported that significantly lower mean serum 25(OH)D levels were found in suicide attempters (*n* = 59) in comparison to non-suicidal depressed patients (*n* = 17) and healthy controls (*n* = 14) [25]. Furthermore, vitamin D status was negatively correlated with one of the inflammatory markers (IL-1 ß) in all subjects as IL-6 was only negatively correlated with vitamin D in non-suicidal depressed patients. In a more recent study by the same research group [28], a significant negative correlation between serum 25(OH)D levels and neutrophile-to-lymphocyte ratio (NLR), white blood cell (WBC) count, as well as inflammation composite score (calculated by summarizing the z-scores of tumor necrosis factor alpha (TNF-α) and IL-6) was found in subjects with major depressive disorder (MDD) and suicidal ideation (*n* = 17), but neither in healthy controls (*n* = 54) nor in patients with MDD without suicidal ideation (*n* = 31). No significant relationship between depression severity and WBC count, NLR, inflammation composite score or 25(OH)D levels was found [28]. Furthermore, MDD status was found to be a significant moderator of the association between serum vitamin D levels and WBC count as well as NLR, but not between inflammation composite score [28]. A recent meta-analysis [29] showed that vitamin D supplementation was followed by a significant reduction in depressive symptoms and CRP levels in patients with mental health problems (schizophrenia, depression, etc.). In summary, many studies indicate the inverse association of CRP, WBC count and IL-6 with vitamin D levels. As Accortt et al. [26] and Shin et al. [27] had reported no relationship between inflammatory markers and depressive symptoms, Grudet et al. [25,28] focused mainly on the vitamin D differences between depressed patients and healthy controls. However, the studies investigating the relationship between depressive symptoms and inflammation reported positive associations [19]. These heterogenous results might be due to methodological differences including examination timeline, different samples and statistical approach.

Overall, the existing body of evidence indicates the complex relationship between depressive symptoms, inflammation and serum vitamin D levels. However, the previous studies were conducted mostly in small samples and with differing methodology. Therefore, the purpose of this study was to examine exploratively whether inflammation acts as a moderator or mediator on the association between vitamin D and depressive symptomatology in a large community sample.

## 2. Materials and Methods

### 2.1. Database and Study Population

We analyzed data obtained during the research project ‘LIFE’ (Leipzig Research Center for Civilisation Diseases). The study population of the LIFE-Adult-Study consisted of 10,000 participants (age range between 18 and 80 years). The recruitment took place between August 2011 and November 2014 in Leipzig, Germany with the detection of the prevalence and incidence of frequent common diseases (sleep disorders, cardiovascular diseases, depression etc.) as one of the main aims. Within the LIFE-Adult-Study several standardized interviews were performed in order to collect medical history and sociodemographic characteristics. Furthermore, physical and medical examinations as well as laboratory tests were conducted. Figure 1 depicts the exclusion process for the current investigation. First, data sets of those LIFE participants who had given a blood sample for the determination of vitamin D and three inflammation markers (see below) were retrieved from the LIFE database (*n* = 9640).

In the first step, datasets were excluded due to containing either non-analyzable serum 25(OH)D levels or missing information on socioeconomic data, body mass index (BMI), as well as missing or non-analyzable scores of Center for Epidemiologic Studies Depression Scale (CES-D). In a second step, participants with certain medical conditions (Parkinson syndrome, Crohn’s disease, liver cirrhosis etc.), medications (corticosteroids for systemic use, oral contraceptives, antineoplastic and immunomodulating agents, anti-infectiva for systemic use, etc.) were excluded, as these were considered to impact the inflammatory markers and serum vitamin D levels, as well as bias the CES-D results. Furthermore, participants with highly dangerous alcohol consumption (>120 g/day for men, >80 g/day for women) or missing information on alcohol and tobacco consumption were excluded. The final data set consisted of 7162 datasets.

### 2.2. Assessment Procedures

#### 2.2.1. Laboratory Assessments

Blood samples from LIFE subjects were collected according to the LIFE study protocol [31].

Measurement of vitamin D total in serum was conducted according to manufacturer’s protocol on an automated laboratory analyzer Cobas 8000 e602 (Roche Diagnostics, Mannheim, Germany) using an electrochemiluminescence immunoassay (ECLIA) with competition principle (Roche Diagnostics Mannheim, Germany). Traceability of the method was standardized against LC-MS/MS [32], and LC-MS/MS in turn was standardized against the NIST standard [33]. The limit of quantitation is 5 ng/mL according to manufacturer information. The primary measurement range is 5–70 ng/mL. Samples with concentration higher than 70 ng/mL were diluted manually 1:2 with Diluent Universal (Roche Diagnostics, Mannheim, Germany). Month of vitamin D measurement was used to define the covariate season as winter (December to February), Spring (March to May), summer (June to August) and autumn (September to November). Leipzig is a city in central Germany. Due to its geographical location (latitude: 51.34°), significant differences in the individual seasons can be observed in the city. During the survey phase of the LIFE study, the average day length in the seasonal periods defined above was [34]: 8:46 h (winter), 13:48 h (spring), 15:42 h (summer) and 10:55 h (fall).

Within the LIFE-adult study, besides blood differential test, only two inflammatory markers have been investigated: interleukin 6 (IL-6) and high-sensitive serum CRP (CRP). IL-6 concentrations were measured in accordance with the manufacturer’s protocol on an automated laboratory analyzer Cobas 8000 e602 (Roche Diagnostics, Mannheim, Germany) with an electrochemiluminescence immunoassay (ECLIA) based on the sandwich principle (Roche Diagnostics Mannheim, Germany). Traceability of this method was standardized against NIBSC (National Institute for Biological Standards and Control) 1st IS 89/548 standard. The limit of quantitation (LoQ) was 1.5 pg/mL. The primary measurement range was 1.5–5000 pg/mL. Samples with concentration >5000 pg/mL were automatically diluted 1:10 up to the maximum measurement range of 50,000 pg/mL.

CRP concentrations were measured in accordance with the manufacturer’s protocol on an automated laboratory analyzer Cobas 8000 c502 (Roche Diagnostics, Mannheim, Germany) with a particle enhanced turbidimetric assay (Roche Diagnostics Mannheim, Germany). The method has been standardized to the IFCC/BCR/CAP reference preparation CRM 470 (RPPHS 91/0619). The lower limit of the assay was 0.15 mg/L. Primary measurement range was 0.15–20 mg/L. Samples with concentration >20 mg/l were automatically diluted 1:15 up to the maximum measurement range of 300 mg/L.

White Blood Cells/Leucocytes (WBC) were measured in EDTA-whole blood with an automated XN-9000 blood analyzer (Sysmex Deutschland GmbH, Norderstedt, Germany).

#### 2.2.2. Questionnaires & Interviews

Within the baseline examination in the LIFE study center, all participants went through a structured interview to assess information on medical history, sociodemographic and socioeconomic data and current medication as well as alcohol and tobacco consumption. A multidimensional index of socioeconomic status was calculated by using the information on education, equivalent household income and occupational status [35,36]. Medical history was assessed by asking for occurrence and treatment of specific diseases. The information on current medication taken in the last week was registered via the ATC (Anatomical Therapeutic Chemical) classification system. 

Anthropometric indices such as body weight and height were also assessed during the study using an electronic scale, which were used to calculate BMI (kg/m^2^). 

The current level of depressive symptomatology within the last week was assessed via the German version of CES-D [30,37]. The CES-D is a structured self-report scale with 20 items. The sum score ranges from 0 to 60, with higher scores pointing to more depressive symptoms. 

### 2.3. Statistical Analyses

Within the final dataset, some participants exhibited values below the detection limit for one or more of the three inflammation markers (see above). To avoid a further reduction of the sample we decided against excluding all those subjects, and performed all moderation and mediation analyses in three largely overlapping but nevertheless different sub-samples: CRP-sample (*n* = 7137), IL-6-sample (*n* = 4026) and WBC-sample (*n* = 7091).

By using descriptive statistics, demographic characteristics, clinical variables and laboratory parameters were analyzed; the choice of these statistics was dependent on the scale level of the corresponding variables. In order to assess the trilateral associations between vitamin D concentrations, inflammation parameters (CRP, IL-6 and WBC) and the intensity of current depressive symptoms (as assessed by using the CES-D sum score) Spearman-Brown correlation coefficients were selected. 

To test the mediation models for the CES-D sum scores as dependent variable, Hayes’ PROCESS macro tool for SPSS for Windows was chosen in order to measure indirect effects. The model was based on a mediation method with 10,000 bootstrap bias-corrected 95% confidence intervals (95% CI). This procedure was in accordance with the recommendations by Hayes [38]. The indirect effect can be interpreted as statistically significant if the afore-mentioned 95% CI does not include the value 0 [39]. Two mediation models were considered: model 1 was without any covariates; in model 2 eight variables (age, gender, socio-economic status, family status, BMI, smoking status, alcohol consumption and the season in which vitamin D concentrations had been measured) were chosen as covariates because of their well-known associations with vitamin serum concentrations, the intensity of depressive symptomatology and/or inflammation parameters. To test moderation effects, we conducted regression analyses which were again either calculated without any covariables (model 1) or with the above mentioned covariables (model 2). In all moderation or mediation models, the independent variable was the serum 25(OH)D concentration, the mediator/moderator was one of the three inflammation parameters (CRP and IL-6 levels or WBC count, respectively) and the dependent variable was the CES-D sum score. 

All statistical analyses were done by using the SPSS software version 26.0 for Windows. Since we investigated the moderation/mediation effects of three different inflammation markers, we used a Bonferroni corrected significance level set at α = 0.05/3 = 0.0167 (two-tailed testing).

## 3. Results

Detailed information regarding characteristics of participants in the total sample as well as the three analyses subsamples is given in Table 1. Approximately 20% of the participants reported mild to severe depressive symptomatology. The mean serum 25(OH)D concentration was 23.31 ng/mL (±11.34 ng/mL), which reflects a vitamin D insufficiency [40].

The trilateral associations between serum 25(OH)D concentrations, inflammation parameters and the CES-D sum scores are summarized in Table 2. Overall, there was a significantly negative correlation between the CES-D and 25(OH)D (rho = −0.07; *p* < 0.001) whereas the association of CES-D and the selected inflammation parameters was significantly positive (rho = 0.094/0.084/0.106; *p* < 0.001). 25(OH)D levels were negatively associated with the inflammation parameters (rho = −0.075/−0.058/−0.077; *p* < 0.001) and all inflammation parameters were characterized by significantly positive intercorrelations (rho = 0.331/0.304/0.231; *p* < 0.001). 

As shown in Figure 2a regarding CRP, in mediation model 1 (without covariates) both the total negative effect of serum 25(OH)D concentrations on the CES-D sum scores (c = −0.0466; t = −6.63; *p* < 0.0001) and the corresponding direct effect (c′ = −0.0452; t = −6.44; *p* < 0.0001) were significant. The negative effect of 25(OH)D on CRP (a = −0.0165; t = −2.98; *p* = 0.0029) was also significant; the same was true for the effect of the CRP on the CES-D scores (b = 0.0831; t = 5.62; *p* < 0.0001). The indirect effect of 25(OH)D on CES-D scores via CRP was non-significant (ab = −0.0014; 98.34% CI: (−0.0034; 0.0002)). In model 2 (with covariates) the latter effect failed to be statistically significant, too (ab = 0.0000; 98.34% CI: (−0.0008; 0.0007); see Figure 2b); moreover, the effect of 25(OH)D concentrations on the CRP concentrations was no longer significant (a = −0.0002; t = −0.04; *p* = 0.9693). The effect of CRP on CES-D scores also failed to meet the Bonferroni corrected level of significance (b = 0.0361; t = 2.39; *p* = 0.0168).

CRP levels were examined as a moderator of the relation between 25(OH) D concentration and CES-D Score (see Table A1). The interaction effect of CRP and 25(OH) D levels did not reach significance neither in model 1 (*p* = 0.0942) nor in model 2 (*p* = 0.2130). Thus, CRP level is no moderator of the relationship between vitamin D and depression.

According to Figure 3a regarding IL-6, both the total effect of 25(OH)D concentrations on CES-D scores (c = −0.0560; t = −6.30; *p* < 0.0001) and the direct effect (c′ = −0.0548; t = −6.16; *p* < 0.0001) were significantly negative in mediation model 1. The negative effect of 25(OH)D on IL-6 failed to reach the Bonferroni corrected level of significance (a = −0.0133; t = −2.28; *p* = 0.0229); while the effect of the IL-6 on CES-D scores was significantly positive (b = 0.0954; t = 4.04; *p* = 0.0001). The indirect effect of 25(OH)D on CES-D scores via IL-6 was not significant (ab = −0.0013; 98.34% CI: (−0.0034; 0.0000)). In mediation model 2 the indirect effect was non-significant, too (ab = −0.0006; 98.34% CI: (−0.0020; 0.0004)) (see Figure 3b). The same was true for the effect of 25(OH)D on IL-6 (a = −0.0097; t = −1.56; *p* = 0.1177).

In the corresponding moderator analyses (see Table A1), the effects of the interaction of the factors 25(OH)D concentration and IL-6 levels on the CES-D Score failed to be significant in model 1 (*p* = 0.1068) as well as model 2 (*p* = 0.2133).

As illustrated by Figure 4a regarding WBC, both the total negative effect of 25(OH)D concentrations on the CES-D scores in mediation model 1 without covariates (c = −0.0470; t = −6.65; *p* < 0.0001) and the corresponding direct effect (c′ = −0.0428; t = −6.06; *p* < 0.0001) were statistically significant. A significantly negative effect of 25(OH)D on WBC was present in this model (a = −0.0116; t = −6.33; *p* < 0.0001); the effect of the WBC on the CES-D scores (b = 0.3644; t = 8.09; *p* < 0.0001) was also significant, as was the indirect effect of 25(OH)D on CES-D scores via WBC (ab = −0.0042; 98.34% CI: (−0.0066; −0.0023)). Similar findings were found for the mediation model 2, with the indirect effect of 25(OH)D on CES-D scores via WBC remaining significant (ab = −0.0011; 98.34% CI: (−0.0025; −0.0002)) (see Figure 4b).

In the respective moderator analyses (see Table A1), the interaction between 25(OH)D and WBC failed to reach the Bonferroni corrected level of significance in model 1 (*p* = 0.0389) and was also non-significant in model 2 (*p* = 0.3794).

## 4. Discussion

To the best of our knowledge, this is the first work examining the role of inflammation in the relation between depressive symptomatology and serum vitamin D levels using mediation and moderation analyses in a community sample. We found negative correlations between vitamin D levels and depressive symptomatology (assessed by CES-D Score) and three inflammatory markers (CRP, IL-6, WBC), which were in turn positively associated with depressive symptomatology.

Results of the mediation models showed a direct effect of vitamin D on depression. Only with regard to WBC could we demonstrate an indirect effect of vitamin D on depressive symptomatology which was partially mediated by inflammation. This finding remained significant when covariables were included in the model but was considerable smaller in size than the direct effect. All of the investigated inflammation markers had effects on depressive symptomatology in the mediation models without covariables. When considering the covariables, only the effect of IL-6 levels and WBC reached significance, with WBC levels again showing the strongest effect. Results of moderation analyses also attested that the association between vitamin D levels and depressive symptomatology was not moderated by any of the investigated inflammatory markers, especially when covariables were included in the analyses.

Our results are mainly in line with previous studies. In all mediation models, vitamin D was negatively associated with CES-D sum scores as demonstrated in most of the previous studies (e.g., [41]). In simple mediator models CRP and WBC count showed significant negative associations with serum 25(OH)D concentrations as reported in the majority of the existing literature (e.g., [42]). However, the association between serum vitamin D levels and IL-6 levels was not significant. There are inconsistent results in previous research regarding this relationship: while some studies reported a significant inverse relationship between serum vitamin D levels and IL-6 (e.g., [43,44]), some demonstrated non-significant results [45,46]. Furthermore, Grudet et al. observed an inverse correlation between vitamin D and Il-6 only in the group of non-suicidal depressed patients (*n* = 17), whereas no significant correlation was found in healthy controls and suicide attempters [25]. Likewise, in another study of this group [28] the correlations between inflammatory markers and serum 25(OH) levels were only significant in participants with MDD and suicidal ideation (*n* = 17), while no significant correlation was found in the healthy control group (*n* = 54) and in participants with MDD but without suicidal ideation (*n* = 31). Besides the differences in study sample and methodology of these studies, these inconsistent results might be due to the role of IL-6’s soluble receptor as the anti-inflammatory mediator, while it is also one of the major pro-inflammatory cytokines for early inflammatory response [47]. 

Positive significant associations between all inflammatory markers and CES-D sum score were found in simple mediator models as expected [24]. However, in mediator models with covariates, only IL-6 levels and WBC count showed a negative association with serum 25(OH)D levels, as the association of serum vitamin D levels and CRP concentrations was no longer significant. Previous studies had demonstrated negative association between CRP and vitamin D levels (e.g., [29,48]) but inconsistent and controversial results were also reported [27], which might be due to methodological differences, since 85.8% of their participants were men. In our analyses we could only demonstrate the mediator effect of WBC on the association between vitamin D levels and depressive symptoms. This might be due to the multiple roles of CRP, IL-6 and WBC in inflammation and their complex interactions with each other and vitamin D. For instance, as CRP is synthesized by vascular and organ-specific cells as well as some types of WBC (lymphocytes and monocytes), it has been shown that IL-6 stimulates the expression of CRP [49]. On the other hand, active vitamin D can suppress proliferation and immunoglobulin production of B lymphocytes (a subtype of white blood cells) [16]. Furthermore, it can modulate the proliferation and function of T lymphocytes, and thereby production of cytokines like IL-6 [16]. Notably, WBC, IL-6 and CRP are accounted as biomarkers both for acute and also systemic chronic inflammation, whereas there are still no specific biomarkers for inflammation [47]. In any case, further research efforts are needed to disentangle and elucidate the complex interactions and interdependencies of the diverse biological processes involved in the inflammatory response.

In moderator analysis none of the three inflammation markers showed a moderator effect on the association between vitamin D and CES-D sum scores. This is contrary to previous results, as Accortt et al. [26] had demonstrated that IL-6 and IL-6/IL-10 ratio moderated the relationship between prenatal serum vitamin D levels and postpartum depressive symptoms. However, it is important to note that the differences in study sample (African American postpartum women vs. the predominantly Caucasian population-based sample in our study) and methodology (assessment time), as well as the role of perinatal biological processes, might explain the opposing results. On the other hand, Grudet et al. [28] examined the moderator effect of MDD status and suicidal ideation on the association of serum vitamin D levels and inflammatory markers. Only the association between serum vitamin D levels and NLR was moderated by MDD status and suicidal ideation.

A strength of the present investigation is that the large study sample consisted of well phenotyped participants of both sexes without serious medical conditions. Apart from a mediation effect of WBC, we found no moderation or mediation effect of inflammation on the association between vitamin D levels and depressive symptomatology, as had been described by previous studies. This could be due to various reasons. First, the differences in sample characteristics including sociodemographic factors might play an important role. Our study population comprised participants without serious medical conditions, while many studies demonstrating a moderation or mediation effect examined clinical populations (postpartum women [26], suicidal and non-suicidal depressives [25,28]). In our study, depressive symptomatology was assessed via self-rating in the CES-D scale, which cannot capture atypical symptoms of depression that might be more relevant for the association between vitamin D levels and depressive symptoms, with a majority of the participants (approximately 80%) achieving clinically inconspicuous scores.

Although various confounding conditions were excluded and several others were included as covariables during the analyses, we cannot rule out that other potential confounders might have influenced the mediation outcomes. Moreover, other inflammatory markers that were found to be relevantly involved in depression such as IL-1 and TNF-α were not available from the LIFE-Adult-Study; for this reason, only IL-6, CRP and WBC were examined in this work. We also excluded datasets in which any of the inflammatory parameters had been labelled as below the respective limit of quantification. While this was rarely the case with respect to CRP levels and WBC count, a rather large proportion of LIFE subjects had undetectable IL-6 levels. Accordingly, almost 2000 data sets had been excluded for the analysis with IL-6. The alternative approach of setting IL-6 values below the detection limit at the fixed value of the detection limit was rejected because of statistical artifacts in the corresponding analyses, which could be attributed to the limited variability due to the many fixed values. Even though the sample for the IL-6 analyses was thus significantly smaller than for the other two immune parameters and a selection bias cannot be completely excluded, the results appear plausible with respect to the direction and strength of the correlation coefficients. Finally, in this work we investigated the mediator and moderator role of inflammation on the association between vitamin D and depressive symptomatology. Nevertheless, a reverse association is also possible. Since some studies suggest that an inflammatory process may reduce vitamin D levels, it can be assumed that inflammation may be linked to depressive symptoms by causing low vitamin D levels. However, studies showing the effect of increasing vitamin D levels on a decrease in most inflammatory markers support, rather, the hypothesis of the role of vitamin D in reducing the inflammation [15,50,51]. Moreover, it could also be assumed that depressive symptoms might affect vitamin D levels by causing sunlight avoiding lifestyle (spending much more time at home, social withdrawal). Furthermore, altered sleep duration caused by depression might lead to an increase in inflammatory parameters. However, a recent meta-analysis has demonstrated that only short sleep duration was associated with increases in IL-6 and CRP [52].

## 5. Conclusions

In conclusion, our work highlights the complex association of vitamin D, inflammation and depressive symptomatology. WBC count partially mediated the association between vitamin D levels and depressive symptomatology, while CRP and IL-6 levels did neither moderate nor mediate this association. Future studies on the role of inflammation on the association of vitamin D and depression are worthwhile and needed. Due to the cross-sectional design of this study, no assumptions regarding causality can be drawn. In the future, longitudinal and prospective studies should examine changes in inflammation and depressive symptoms, as well as the effect of vitamin D supplementation on inflammation and depressive symptomatology for causality assessment. This might help to shed a new light on the potential role of vitamin D supplementation as an anti-inflammatory therapy option for depression among individuals with low vitamin D levels. 

## Figures and Tables

**Figure 1 nutrients-13-01972-f001:**
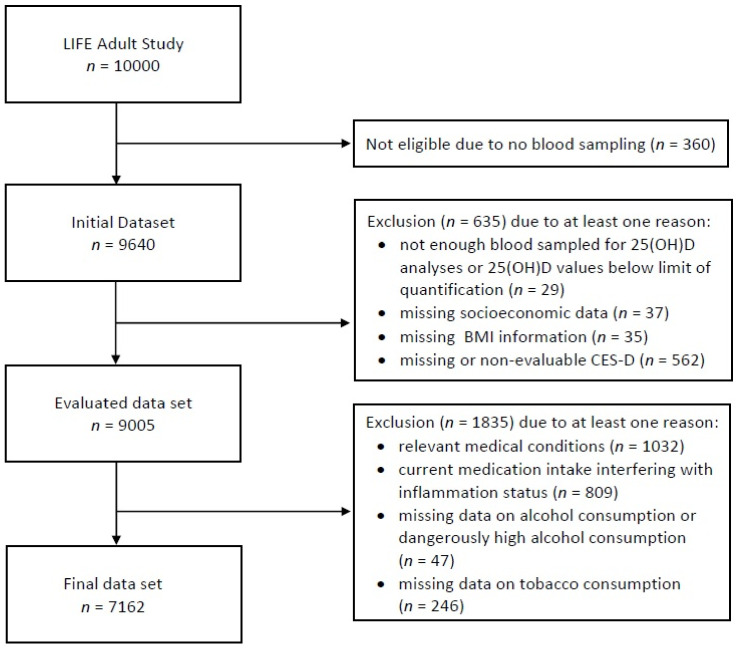
Flow diagram of data selection process. BMI: body mass index, CES-D: German version of the Centre for Epidemiological Studies Depression Scale [30], 25(OH)D: 25-Hydroxyvitamin D.

**Figure 2 nutrients-13-01972-f002:**
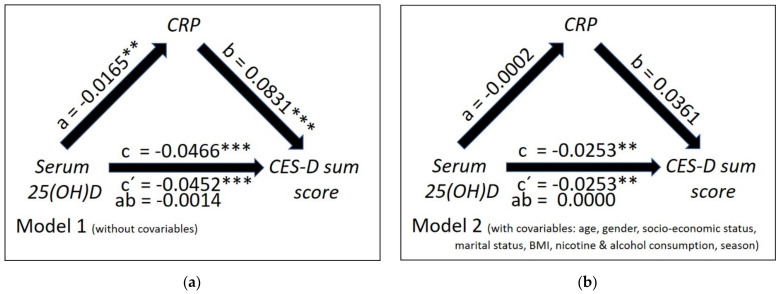
Mediation model without (**a**) and with covariates (**b**) showing both the total and direct effect of serum 25(OH)D concentrations on the intensity of depressive symptomatology as measured by CES-D sum scores (path coefficients c and c′) and the indirect effect of serum 25(OH)D concentrations on CES-D scores mediated through c-reactive protein (CRP) concentrations (path coefficient ab) in the study sample. The figures depict the unstandardized path coefficients (a,b,c,c′ and ab; with ** *p* < 0.010; *** *p* < 0.001).

**Figure 3 nutrients-13-01972-f003:**
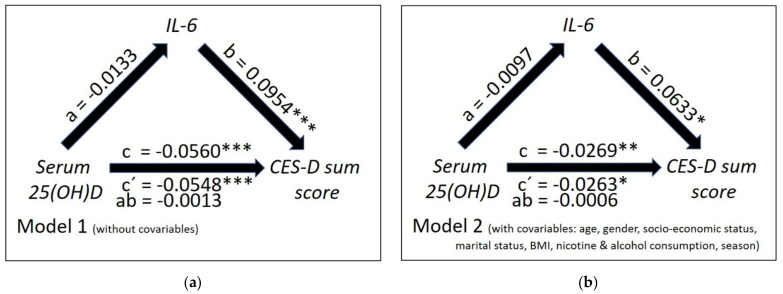
Mediation model without (**a**) and with covariates (**b**) showing both the total and direct effect of serum 25(OH)D concentrations on the intensity of depressive symptomatology as measured by CES-D sum scores (path coefficients c and c′) and the indirect effect of serum 25(OH)D concentrations on CES-D scores mediated through interleukin 6 (IL-6) concentrations (path coefficient ab) in the study sample. The figure depicts the unstandardized path coefficients (a,b,c,c′ and ab; with * *p* < 0.050; ** *p* < 0.010; *** *p* < 0.001).

**Figure 4 nutrients-13-01972-f004:**
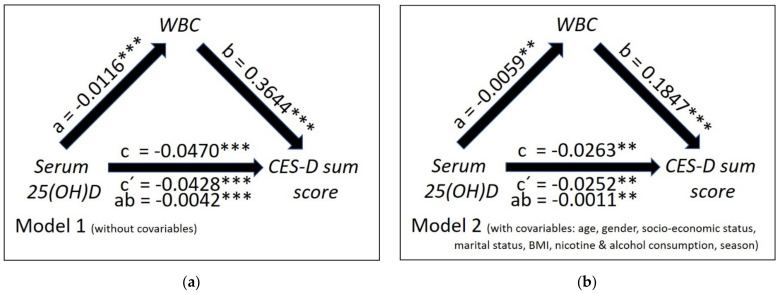
Mediation model without (**a**) and with covariates (**b**) showing both the total and direct effect of serum 25(OH)D concentrations on the intensity of depressive symptomatology as measured by CES-D sum scores (path coefficients c and c′) and the indirect effect of serum 25(OH)D concentrations on CES-D scores mediated through white blood cell (WBC) count (path coefficient ab) in the study sample. The figure depicts the unstandardized path coefficients (a,b,c,c′ and ab; with ** *p* < 0.010; *** *p* < 0.001).

**Table 1 nutrients-13-01972-t001:** Characteristics of the final sample as well as three analysis subsamples.

Variables	Total Sample (*n* = 7162)	CRP Sample ^1^ (*n* = 7137)	IL-6 Sample ^2^ (*n* = 4026)	WBC Sample ^3^ (*n* = 7091)
Sex, *n* (%)				
Females	3420 (47.8%)	3409 (47.8%)	1860 (46.2%)	3386 (47.8%)
Males	3742 (52.2%)	3728 (52.2%)	2166 (53.6%)	3705 (52.2%)
Age (years), mean (SD)	56.48 (11.94)	56.52 (11.93)	57.11 (12.03)	56.49 (11.94)
Age groups, *n* (%)				
18–39 years	342 (4.8%)	338 (4.7%)	190 (4.7%)	339 (4.8%)
40–49 years	1929 (26.9%)	1916 (26.8%)	1034 (25.7%)	1911 (26.9%)
50–59 years	1828 (25.5%)	1824 (25.6%)	976 (24.2%)	1807 (25.5%)
60–69 years	1831 (25.6%)	1829 (25.6%)	1058 (26.3%)	1813 (25.6%)
70+ years	1232 (17.2%)	1230 (17.2%)	768 (19.1%)	1221 (17.2%)
Family status, *n* (%)				
Married, living together	4369 (60.9%)	4360 (61.1%)	2406 (59.8%)	4325 (61.0%)
Single	1270 (17.8%)	1256 (17.5%)	730 (18.1%)	1256 (17.7%)
Others (divorced, widowed, living separated)	1523 (21.3%)	1521 (21.3%)	890 (22.1%)	1510 (21.3%)
Socioeconomic status, *n* (%)				
High	1525 (21.3%)	1519 (21.3%)	829 (20.6%)	1507 (21.2%)
Middle	4333 (60.5%)	4319 (60.5%)	2406 (59.8%)	4292 (60.5%)
Low	1304 (18.2%)	1299 (18.2%)	791 (19.6%)	1292 (18.2%)
Employment status, *n* (%)				
Gainful employment	4028 (56.2%)	4008 (56.2%)	2148 (53.4%)	3980 (56.1%)
Unemployment	650 (9.1%)	649 (9.1%)	379 (9.4%)	645 (9.1%)
Retirement	2484 (34.7%)	2480 (34.7%)	1499 (37.2%)	2466 (34.8%)
BMI (kg/m^2^), mean (SD)	27.37 (4.82)	27.39 (4.82)	27.84 (5.09)	27.36 (4.80)
BMI categories, *n* (%)				
Underweight (<18.5)	31 (0.4%)	29 (0.4%)	16 (0.4%)	31 (0.4%)
Normal wight (18.5–24.9)	2420 (33.8%)	2403 (33.7%)	1254 (31.1%)	2397 (33.8%)
Overweight (25.0–29.9)	2948 (41.2%)	2943 (41.2%)	1603 (39.2%)	2922 (41.2%)
Obesity (≥30)	1763 (24.6%)	1762 (24.7%)	1153 (28.6%)	1741 (24.6%)
Consumption of alcohol (g/day), mean (SD)	12.95 (17.95)	12.95 (17.96)	13.53 (19.05)	12.98 (17.95)
Smoking status, *n* (%)				
Current smoker	1628 (22.7%)	1624 (22.8%)	989 (24.6%)	1614 (22.8%)
Former smoker	2096 (29.3%)	2090 (29.3%)	1175 (29.2%)	2076 (29.3%)
Non-smoker	3438 (48.0%)	3423 (48.0%)	1862 (46.2%)	3401 (48.0%)
Season, *n* (%)				
Winter	1611 (22.5%)	1610 (22.6%)	930 (23.1%)	1591 (22.94%)
Spring	1840 (25.7%)	1839 (25.8%)	778 (19.3%)	1823 (25.7%)
Summer	1871 (26.1%)	1860 (26.1%)	1214 (30.4%)	1858 (26.2%)
Autumn	1840 (25.7%)	1828 (25.6%)	1104 (27.04%)	1819 (25.7%)
Serum 25(OH)D concentration (ng/mL), mean (SD)	23.34 (11.33)	23.32 (11.32)	24.41 (11.95)	23.34 (11.32)
CES-D ^4^ Sum Score, mean (SD)	10.54 (6.81)	10.53 (6.81)	10.67 (6.84)	10.54 (6.83)
CES-D ^4^ severity, *n* (%)				
0–14 (clinically inconspicuous)	5703 (79.6%)	5685 (79.7%)	3174 (78.8%)	5644 (79.6%)
15–21 (mild to moderate)	987 (13.8%)	983 (13.8%)	595 (14.8%)	979 (13.9%)
≥22 (severe)	472 (6.6%)	469 (6.6%)	257 (6.4%)	468 (6.6%)

^1^ CRP-Sample: all participants with analyzable information on c-reactive protein levels. ^2^ IL6-Sample: all participants with analyzable information on interleukine-6 levels. ^3^ WBC-Sample: all participants with analyzable information on white blood cell counts. ^4^ CES-D: German version of the Centre for Epidemiological Studies Depression Scale ([37], German version by [30]).

**Table 2 nutrients-13-01972-t002:** Correlation coefficient matrix of serum 25(OH)D concentrations, CES-D sum scores and inflammatory markers.

	CES-D ^1^ Sum Score	25(OH)D ^2^ Concentration	CRP ^3^ Levels	IL-6 ^4^ Levels	WBC ^5^ Count
CES-D^1^ sum score	1	−0.074	0.094	0.084	0.106
		*p* < 0.001	*p* < 0.001	*p* < 0.001	*p* < 0.001
	*n* = 7162	*n* = 7162	*n* = 7145	*n* = 4028	*n* = 7099
Serum 25(OH)D ^2^ concentration		1	−0.075	−0.058	−0.077
			*p* < 0.001	*p* < 0.001	*p* < 0.001
		*n* = 7162	*n* = 7137	*n* = 4026	*n* = 7091
CRP ^3^ levels			1	0.331	0.304
				*p* < 0.001	*p* < 0.001
			*n* = 7145	*n*=4009	*n* = 7076
IL6 ^4^ levels				1	0.231
				*p* < 0.001	*p* < 0.001
				*n* = 4028	*n* = 3988
WBC ^5^ count					1

					*n* = 7099

^1^ CES-D: German version of the Center for Epidemiological Studies Depression Scale; ^2.^ 25(OH)D: 25-Hydroxyvitamin D; ^3^ CRP: C-reactive protein; ^4^ IL-6: Interleukin 6, ^5^ WBC: White blood cells. Spearman-Brown correlation coefficients are given.

## Data Availability

Restrictions apply to the availability of these data. Restrictions apply to the availability of these data. Data was obtained from the Leipzig Research Center for Civilisation Diseases. All data and samples of LIFE are the property of the University of Leipzig and are subject to the Law for the Protection of Informal Self-Determination in the Free State of Saxony (Saxon Data Protection Act). Use of data can be requested through the LIFE office (https://life.uni-leipzig.de/ accessed on 7 June 2021).

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
