# Peer review of "Inflammation and the Association of Vitamin D and Depressive Symptomatology"

_nutrients, 2021, doi:10.3390/nu13061972_

Round 1

Reviewer 1 Report

The purpose of this study was to examine exploratively whether inflammation acts as a moderator or mediator on the association between vitamin D and depressive symptomatology in a large community sample.

Though the draft is carefully written and it is an important topic, I have several issues to be addressed.

Major point

Introduction

  1. L78-L108
    If the novelty of your paper is the large size of the study population, could you please add the sample number of each study in the paragraph?
  2. 109-
    To make the context more understandable, could you summarize the association of each inflammation markers and vitamin D deficiency and depression in the last paragraph of the introduction part?

Materials and Method

  1. 117-139
    Since you used “season” as a covariate, could you add a bit more about the background of Leipzig, especially about the daylight hours in each season?
  2. L145-
    Please add the explanation why you chose those 3 inflammatory markers (IL-6, WBC, CRP).

Statistical Analysis

  1. Why don’t you use multiple imputations to fully utilize your data for the three analysis for the difference inflammation markers?

Discussion

  1. 368-380

Can you add the discussion why only WBC showed the mediation effect in the study from the viewpoint of the characteristics of each inflammatory marker, i.e. chronic or acute?

Minor point

L116, 144

The titles of the section are both “Database and Study Population” for both 2.1 and 2.2.

Author Response

Reviewer 1

Major point

Introduction

  1. L78-L108
    If the novelty of your paper is the large size of the study population, could you please add the sample number of each study in the paragraph?

Response: Thank you for your feedback. We have added the sample sizes as suggested (lines 81-103).

  1. 109-
    To make the context more understandable, could you summarize the association of each inflammation markers and vitamin D deficiency and depression in the last paragraph of the introduction part?

Response: Thank you for this suggestion. We have extended the fourth paragraph (lines 110-117) to supply the requested summary of the association between those parameters. However, it was the result of our initial literature research that, as of now, there are rather heterogeneous research results. Therefore, we wanted our study to contribute to the existing body of knowledge and add to it with a larger and more balanced study population.

Materials and Method

  1. 117-139
    Since you used “season” as a covariate, could you add a bit more about the background of Leipzig, especially about the daylight hours in each season?

Response: As suggested by the reviewer, we have added a few more information about the geographic location of Leipzig and the mean duration of sunlight hours in the seasonal periods during the duration of the LIFE-study (see lines 165-171).

  1. L145-
    Please add the explanation why you chose those 3 inflammatory markers (IL-6, WBC, CRP).

Response: We used data from the LIFE-Adult study. Within this project, besides blood differential test, only the two inflammatory markers IL-6 and CRP have been investigated. Had more inflammation parameters been available, we would have included them in our analyses. We have added an explanation in the materials and methods (lines 173/174).

Statistical Analysis

  1. Why don’t you use multiple imputations to fully utilize your data for the three analysis for the difference inflammation markers?

Response: The reviewer raises a point that we also thought about extensively when preparing the article. We abstained from multiple imputation because the proportion of missing values (about 30%) was rather high and the missing values were not at random. Missing values of inflammatory parameters were dependent from the parameter itself since apart from a select few cases where missing values were due to blood samples that could not be evaluated, data was missing if the values were extremely low and thus below the threshold for detection. Thus, an important criterion for the application of multiple imputation - presence of missings at random - was not fulfilled.

Discussion

  1. 368-380
    Can you add the discussion why only WBC showed the mediation effect in the study from the viewpoint of the characteristics of each inflammatory marker, i.e. chronic or acute?

Response: Thank you for your feedback. We have elaborated on this aspect in the discussion (see lines 383-404). However, there is need for further investigation on this topic.

Minor point

L116, 144
The titles of the section are both “Database and Study Population” for both 2.1 and 2.2.

Response: Thank you very much for your remark. A slight error must have crept in while adjusting the layout. We have changed the title of 2.2 to “Assessment procedures”.

Reviewer 2 Report

This study is well done and the conclusion sounds well taken. A strength is the fine study design and the high number of investigated persons.

Nevertheless, I have one minor criticism :

Depressive persons tend to retract themsselves with longer stays at home, and also longer sleeping durations. These factors may lower the active vitamin D status by withdrawal of sunlight. 

The authors  did not consider this fact. I recommend to discuss this under the section "limitations".

Author Response

Reviewer 2

Depressive persons tend to retract themselves with longer stays at home, and also longer sleeping durations. These factors may lower the active vitamin D status by withdrawal of sunlight. 

The authors did not consider this fact. I recommend to discuss this under the section "limitations".

Response: Thank you very much for your feedback and recommendation. We have discussed this point under the section “limitations” (see lines 453-458).

Round 2

Reviewer 1 Report

Thank you for the careful revision. I don't have any additional comments.